# A linear time approximation of Wasserstein distance with word embedding selection

**Sho Otao**[1] and **Makoto Yamada**[2]
[1]Kyoto University, Japan
[2]Okinawa Institute of Science and Technology, Japan
sho.otao@ml.ist.i.kyoto-u.ac.jp, makoto.yamada@oist.jp

## Abstract

Wasserstein distance, which can be computed by solving the optimal transport problem, is a powerful method for measuring the dissimilarity between documents. In the NLP community, it is referred to as word mover's distance (WMD). One of the key challenges of Wasserstein distance is its computational cost since it needs cubic time. Although the Sinkhorn algorithm is a powerful tool to speed up to compute the Wasserstein distance, it still requires square time. Recently, a linear time approximation of the Wasserstein distance including the sliced Wasserstein and the tree-Wasserstein distance (TWD) has been proposed. However, a linear time approximation method suffers when the dimensionality of word vectors is high. In this study, we propose a method to combine feature selection and tree approximation of Wasserstein distance to handle high-dimensional problems. More specifically, we use multiple word embeddings and automatically select useful word embeddings in a tree approximation of Wasserstein distance. To this end, we approximate Wasserstein distance for each word vector by tree approximation technique, and select the discriminative (i.e., large Wasserstein distance) word embeddings by solving an entropic regularized maximization problem. Through our experiments on document classification, our proposed method achieved high performance.

## 1 Introduction

Wasserstein distance, which is obtained by solving the optimal transport problem, is a powerful method for measuring the distance between distributions and is used in natural language processing to measure dissimilarity between documents (Kusner et al., 2015; Huang et al., 2016; Yurochkin et al., 2019; Yokoi et al., 2020; Sato et al., 2022). Kusner et al. (2015) proposed word mover's distance (WMD), which uses word vectors to compute the ground metric of the Wasserstein distance and can measure dissimilarity between documents using Wasserstein distance. More specifically, given a word embedding and the frequency of the words in each document, WMD is formulated as an optimal transport problem in the word embedding space.

Although the optimal transport problem can be solved using a linear programming algorithm, it requires cubic computational time with respect to the number of samples. Cuturi (2013) proposed to introduce the entropy regularization to the optimal transport problem, which can be solved with a square computational time with respect to the number of samples by the Sinkhorn algorithm. However, the Sinkhorn algorithm is too slow for practical application in tasks such as natural language processing, that involve a large number of words. The tree-Wasserstein distance (TWD) is a method to measure the distance between distributions using a tree structure, which can approximate Wasserstein distance with a linear computational time with respect to the number of nodes in the tree (Indyk and Thaper, 2003; Le et al., 2019; Backurs et al., 2020; Sato et al., 2020; Takezawa et al., 2021, 2022; Yamada et al., 2022; Chen et al., 2022). As a method for constructing tree structures, Indyk and Thaper (2003) proposed Quadtree, which constructs a tree by recursively partitioning the original space. In addition, Le et al. (2019) used hierarchical clustering to construct a tree. Existing approaches successfully approximate the original optimal transport problems. However, when the dimensionality of the vectors (e.g., word vectors) is high, their performance can be degraded due to the curse of dimensionality.

To address high-dimensional problems, robust variants of optimal transport for high-dimensional data have recently been proposed (Paty and Cuturi, 2019; Petrovich et al., 2022). Paty and Cuturi (2019) proposed the subspace robust Wasserstein distances (SRW), which projects data into a subspace and considers optimal transport in the subspace. Petrovich et al. (2022) proposed feature-

robust optimal transport (FROT), which measures distances between distributions with group feature selection when feature groups are given as prior information. Since SRW and FROT alternately solve the feature selection optimization problems and the optimal transport problem with either linear programming or the Sinkhorn algorithm, they do not scale to large datasets. Therefore, applying these techniques to NLP tasks such as document classification that involve large amounts of data is difficult.

In this study, we propose the tree-Wasserstein distance with group feature selection (TWD-GFS) to combine feature selection and TWD to handle high-dimensional problems and scale to large datasets. We consider a setting in which some distributions are given as training data to select discriminative feature groups in advance. TWD-GFS first constructs a tree structure for each of the given feature groups (i.e., word embeddings), and its output is the weighted sum of the TWD calculated using each tree. The weight for each feature group can be obtained by solving an entropic regularized maximization problem using the distribution of the training data. After training, TWD-GFS can measure distances with selected features much faster than SRW and FROT. Through the synthetic experiments, we show that our proposed method can select discriminative features in a high-dimensional setting with a large number of noise features. Then, through the real experiment, we consider multiple word embeddings as feature groups and confirmed the performance of TWD-GFS on word embedding-based document classification tasks with word embedding selection.

**Contributions:** Our contributions are as follows:

- We propose a fast TWD-based method to select discriminative feature groups and measure distances between distributions.

- Through synthetic experiments, we show that our proposed method can select discriminative features in a high-dimensional setting with a large number of noise features.

- Through real experiments with document classification tasks, we consider multiple word embeddings as a feature group and show our proposed method with word embedding selection achieved high performance.

**Notation:** In the following sections, we write $\mathbf{1}_n$

for a $n$-dimensional vector with all ones, and $\Sigma^n$ for a $n$-dimensional probability simplex.

## 2 Related works

In this section, we introduce Wasserstein distance, tree-Wasserstein distance, and optimal transport with feature selection.

### 2.1 Wasserstein distance

Given two set of $d$-dimensional vectors in euclidean spaces $\mathbf{X} = (\mathbf{x}_1, ..., \mathbf{x}_n) \in \mathbb{R}^{d \times n}, \mathbf{Y} = (\mathbf{y}_1, ..., \mathbf{y}_m) \in \mathbb{R}^{d \times m}$ and two weights $\mathbf{a} \in \Sigma^n, \mathbf{b} \in \Sigma^m$, we consider two discrete distributions $\mu = \sum_{i=1}^{n} a_i \delta_{\mathbf{x}_i}, \nu = \sum_{j=1}^{m} b_j \delta_{\mathbf{y}_j}$. The p-Wasserstein distance, which measures the distance between $\mu$ and $\nu$, is defined as follows:

$$W_p(\mu, \nu) = \left( \min_{\mathbf{\Pi} \in \mathbf{U}(\mu, \nu)} \sum_{i=1}^{n} \sum_{j=1}^{m} \pi_{ij} c(\mathbf{x}_i, \mathbf{y}_j)^p \right)^{\frac{1}{p}},$$

where $c(\mathbf{x}_i, \mathbf{y}_j)$ is a cost between $\mathbf{x}_i$ and $\mathbf{y}_j$, $\mathbf{\Pi}$ is the transport plan, and $\mathbf{U}(\mu, \nu)$ is the following set of transport plans.

$$\mathbf{U}(\mu, \nu) = \{\mathbf{\Pi} \in \mathbb{R}_+^{n \times m} : \mathbf{\Pi} \mathbf{1}_m = \mathbf{a}, \mathbf{\Pi}^\top \mathbf{1}_n = \mathbf{b}\}.$$

This optimization problem is referred to as the optimal transport problem (Villani, 2009; Peyré and Cuturi, 2019). The optimal transport problem can be solved using a linear programming algorithm, however, it requires $O(n^3)(n = m)$ computational time. Cuturi (2013) introduced entropy regularization into the optimal transport problem, which can be solved in $O(nm)$ computational time using the Sinkhorn algorithm.

### 2.2 Tree-Wasserstein distance

Tree-Wasserstein distance (TWD) (Indyk and Thaper, 2003; Le et al., 2019) approximates the 1-Wasserstein distance on a tree structure in a linear time for the number of nodes in the tree. Given a tree $\mathcal{T} = (V, E)$ where $V$ and $E$ are the sets of nodes and edges, and a tree metric $d_\mathcal{T}$, the TWD between two distributions $\mu$ and $\nu$ supported by $\mathcal{T}$ is defined as follows:

$$W_{d_\mathcal{T}}(\mu, \nu) = \sum_{e \in E} w_e |\mu(\Gamma(v_e)) - \nu(\Gamma(v_e))|,$$

where $e$ is an edge, $w_e \in \mathbb{R}_+$ is a weight of the edge $e$, $v_e$ is a node on the far side from the root of edge $e$, and $\Gamma(v_e)$ is a set of nodes belonging to a

subtree whose root is node $v_e$. Moreover, Takezawa et al. (2021) proposed a matrix-form formulation of TWD.

Because TWD is computed using a given tree, the tree metric must represent the metric of the original space well. The tree metric is determined by a tree structure and edge weights, and several methods to construct them have been proposed. First, Indyk and Thaper (2003) proposed Quadtree, which is constructed by recursively dividing an original space into $2^d$ regions, and the edge weights are $2^{-l(e)}$ where $l(e)$ is the depth of the edge $e$. Subsequently, Le et al. (2019) proposed an algorithm based on clustering. This tree is constructed by recursively clustering data with farthest-point clustering, and the edge weights are the distances between the cluster centers in the original space. We call this algorithm Clustertree.

Initial values must be provided for the center point of the split in Quadtree and the cluster centers in Clustertree, so they involve the disadvantage that the randomness of these initializations can significantly change the tree structure. Tree-sliced Wasserstein distance (TSW) (Le et al., 2019), which uses multiple trees constructed based on different initializations, is defined as follows:

$$TSW(\mu, \nu) = \frac{1}{T} \sum_{t=1}^{T} W_{d_{\mathcal{T}^{(t)}}}(\mu, \nu),$$

where $W_{d_{\mathcal{T}^{(t)}}}(\mu, \nu)$ is TWD calculated on a tree $t$, and $T$ is the number of trees. TSW can mitigate the effects of these initialization problems.

### 2.3 Optimal transport with feature selection

Wasserstein distance is highly dependent on a cost function $c$. Therefore, discriminative features with large distances between points are useful for calculating the Wasserstein distance. Based on this idea, robust variants of optimal transport for high-dimensional data, which measures distances between distributions while selecting discriminative features, have been proposed in recent years (Paty and Cuturi, 2019; Petrovich et al., 2022). These methods select discriminative features that maximize the distance of distributions and measure distances in a selected feature space. Paty and Cuturi (2019) proposed subspace robust Wasserstein distances (SRW), which projects data into a discriminative subspace and computes optimal transport problem in the subspace. Also, Petrovich et al. (2022) proposed feature-robust optimal transport (FROT), which measures distances based on selected feature groups when the information of feature groups is given.

## 3 Proposed method

In this section, we describe a problem setting considered in this research, our proposed method and a comparison between our approach and existing techniques.

### 3.1 Problem settings

We suppose that the features of $\mathbf{x} \in \mathbb{R}^d$ are separated by $F$ groups $\mathbf{x} = (\mathbf{x}^{(1)^\top}, ..., \mathbf{x}^{(F)^\top})^\top$ where $\mathbf{x}^{(f)} \in \mathbb{R}^{d_f}$ and $\sum_{f=1}^{F} d_f = d$. Even if no prior information on feature groups is given, each feature can be considered as a single group. We write the set of $N$ vectors used in constructing the tree as $\mathbf{X} = (\mathbf{x}_1, ..., \mathbf{x}_N) \in \mathbb{R}^{d \times N}$. In document classification tasks, $\mathbf{X}$ is a set of word vectors of all words in a dataset. We can rewrite $\mathbf{X}$ as $(\mathbf{X}^{(1)^\top}, ..., \mathbf{X}^{(F)^\top})^\top$, where $\mathbf{X}^{(f)} = (\mathbf{x}_1^{(f)}, ..., \mathbf{x}_N^{(f)}) \in \mathbb{R}^{d_f \times N}$. For example, $\mathbf{x}_i^{(1)}$ is a word2vec, and $\mathbf{x}_i^{(2)}$ is a Glove vector when considering multiple word embeddings as feature groups.

In this study, we consider a setting in which some distributions (i.e., documents) $\{\mu_i\}_{i=1}^{M}$ are given as training data for group feature selection. $M$ is the number of available distributions. This setting is natural for the experiments with the $k$-nearest neighbor method in Section 5. Note that the labels of distributions are not given.

### 3.2 Tree-Wasserstein distance with group feature selection

For high-dimensional vectors, the performance of TWD can be degraded due to the curse of dimensionality. The existing feature-robust optimal transport methods SRW (Paty and Cuturi, 2019) and FROT (Petrovich et al., 2022) alternately solve the optimal transport problem with a high cost and the feature selection optimization problem. Hence, they cannot scale to large datasets which involve large amounts of words. Then, we propose tree-Wasserstein distance with group feature selection (TWD-GFS) to combine feature selection and TWD to handle high-dimensional problems and scale to large datasets.

TWD-GFS constructs $F$ trees using each feature group $\mathbf{X}^{(f)}(1 \leq f \leq F)$, and then outputs the weighted sum of $F$ TWDs with learned weights

$\mathbf{w}^* \in \Sigma^F$ as the distance of distributions. TWD-GFS between $\mu$ and $\nu$ is defined as follows:

$$\sum_{f=1}^{F} w_f^* W_{d_{\mathcal{T}_f}}(\mu, \nu), \tag{1}$$

where $\mathcal{T}_f$ is the tree constructed by using the feature group $f$ and $W_{d_{\mathcal{T}_f}}(\mu, \nu)$ is TWD between two distributions $\mu, \nu$ using $\mathcal{T}_f$.

As well as SRW (Paty and Cuturi, 2019) and FROT (Petrovich et al., 2022), we learn weights to maximize (1), distances between distributions. The learned parameter $\mathbf{w}^*$ requires that the weight $w_f$ corresponding to the discriminative feature group $f$ be larger. We create an index-pair set $\Omega$ from the available distributions $\{\mu_i\}_{i=1}^{M}$, and learn $\mathbf{w}$ by solving the following problem:

$$\mathbf{w}^* = \arg\max_{\mathbf{w} \in \Sigma^F} \sum_{(i,j) \in \Omega} \left\{ \sum_{f=1}^{F} w_f W_{d_{\mathcal{T}_f}}(\mu_i, \mu_j) \right\}. \tag{2}$$

To simplify the notation, we define $\boldsymbol{\phi}_\Omega \in \mathbb{R}^F$ as follows:

$$\boldsymbol{\phi}_\Omega = \begin{pmatrix} \sum_{(i,j) \in \Omega} W_{d_{\mathcal{T}_1}}(\mu_i, \mu_j) \\ \vdots \\ \sum_{(i,j) \in \Omega} W_{d_{\mathcal{T}_F}}(\mu_i, \mu_j) \end{pmatrix},$$

where $f$th dimension of $\boldsymbol{\phi}_\Omega$ is the sum of TWDs calculated for $\Omega$ using the tree $\mathcal{T}_f$. By using $\boldsymbol{\phi}_\Omega$, we can rewrite (2) as follows:

$$\mathbf{w}^* = \arg\max_{\mathbf{w} \in \Sigma^F} \mathbf{w}^\top \boldsymbol{\phi}_\Omega,$$

where $\mathbf{w}^*$ is a one-hot vector with a value of 1 for one dimension with the largest value in $\boldsymbol{\phi}_\Omega$ and 0 for the other dimensions. However, we sometimes want more than one useful feature group. Then, we introduce the entropic regularization $H(\mathbf{w}) = -\sum_{f=1}^{F} w_f(\log w_f - 1)$ and consider the following problem such that $\mathbf{w}^*$ is no longer a one-hot vector:

$$\mathbf{w}^* = \arg\max_{\mathbf{w} \in \mathbb{R}^F} \mathbf{w}^\top \boldsymbol{\phi}_\Omega + \eta H(\mathbf{w}) \tag{3}$$
$$\text{s.t.} \quad \mathbf{w}^\top \mathbf{1}_F = 1,$$

where $\eta \geq 0$ is a hyperparameter. The smaller $\eta$, the closer $\mathbf{w}^*$ is to a one-hot vector, and the larger $\eta$, the closer $\mathbf{w}^*$ is to a vector where all values are $1/F$. Note that $H(\mathbf{w})$ naturally satisfies the nonnegative constraint. Since $H(\mathbf{w})$ is a strongly

---

**Algorithm 1** Weight learning algorithm.

**Input:** The set of points separated by feature groups $\mathbf{X} = (\mathbf{X}^{(1)^\top}, ..., \mathbf{X}^{(F)^\top})^\top \in \mathbb{R}^{d \times N}$, the training index-pair set $\Omega$, the hyperparameter $\eta \in \mathbb{R}$.

**Output:** $\mathbf{w}^* \in \mathbb{R}^F$

1: Construct $(\mathcal{T}_1, ..., \mathcal{T}_F)$ by using each feature group.

2: Calculate TWD for $\Omega$ using each tree and make
$$\boldsymbol{\phi}_\Omega = \begin{pmatrix} \sum_{(i,j) \in \Omega} W_{d_{\mathcal{T}_1}}(\mu_i, \mu_j) \\ \vdots \\ \sum_{(i,j) \in \Omega} W_{d_{\mathcal{T}_F}}(\mu_i, \mu_j) \end{pmatrix}.$$

3: Set $w_f^*$ as $\dfrac{\exp\left(\frac{(\boldsymbol{\phi}_\Omega)_f}{\eta}\right)}{\sum_{k=1}^{F} \exp\left(\frac{(\boldsymbol{\phi}_\Omega)_k}{\eta}\right)} (1 \leq f \leq F)$.

---

concave function, the constrained optimization of (3) is a concave maximization and the optimal solution of (3) is uniquely obtained as follows:

$$w_f^* = \frac{\exp\left(\frac{(\boldsymbol{\phi}_\Omega)_f}{\eta}\right)}{\sum_{k=1}^{F} \exp\left(\frac{(\boldsymbol{\phi}_\Omega)_k}{\eta}\right)}. \tag{4}$$

This derivation is given in Appendix A.

Given the points $\mathbf{X} = (\mathbf{X}^{(1)^\top}, ..., \mathbf{X}^{(F)^\top})^\top \in \mathbb{R}^{d \times N}$, the index-pair set $\Omega$, and the hyperparameter $\eta \in \mathbb{R}$, we obtain $\mathbf{w}^*$ according to Algorithm 1. After learning $\mathbf{w}^*$, we measure the distances of the distributions by (1) using $F$ trees constructed during training.

### 3.3 Comparison with other methods

In this subsection, we compare our proposed approach with existing methods from two points of view.

**Optimal transport with feature selection** SRW (Paty and Cuturi, 2019) and FROT (Petrovich et al., 2022) do not select features in advance and instead optimize for feature selection while solving the optimal transport problem, which increases their computational time. TWD-GFS selects feature groups using some distributions in advance, so no additional optimization is required after training. Therefore, TWD-GFS can measure distances between distributions much faster than SRW and FROT after training. The training time of TWD-GFS depends on the time required to compute TWD, so it is fast and does not have a high cost. In addition, TWD-GFS uses multiple distributions as training data,

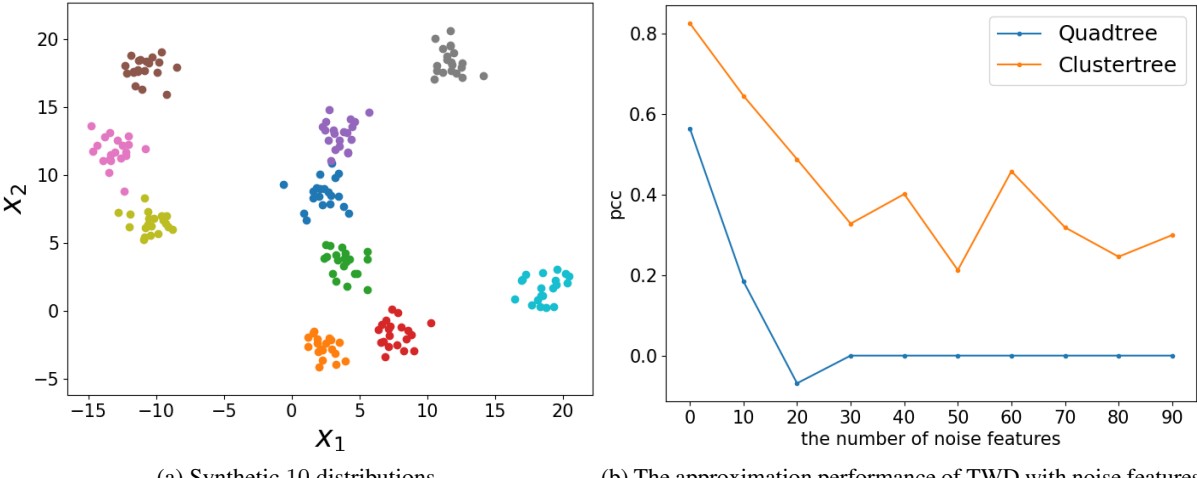

(a) Synthetic 10 distributions.

(b) The approximation performance of TWD with noise features.

Figure 1: The approximation performance of TWD with added noise features for synthetic 10 distributions.

which has the advantage of selecting useful feature groups in common across the entire dataset compared to SRW and FROT.

**TWD with learning tree structures and edge weights**  TWD-GFS learns only tree weights and uses Quadtree and Clustertree as trees. Several methods have been proposed to learn tree structures and edge weights (Takezawa et al., 2021; Yamada et al., 2022; Chen et al., 2022), and the problem setting used in these methods differs from that of our proposed approach. First, Takezawa et al. (2021) proposed a method to learn task-specific tree structures when the distributions and the labels of the distributions are given as training data. Because we consider the settings where the labels of the distributions are not available, this method cannot be applied in our problem setting. Subsequently, Yamada et al. (2022) and Chen et al. (2022) proposed the methods to learn edge weights or tree structures and approximate the 1-Wasserstein distance accurately. These methods use all features, making it difficult to apply them to our settings in which the original data contains unnecessary features. In future research, we expect these methods to be combined with feature selection.

## 4   Synthetic experiments

In this section, we show that our proposed method can select the correct feature groups in high-dimensional settings with a lot of noise features.

In this experiment, we first created 10 distributions in two dimensions as shown in Figure 1a. Each distribution contained 20 points and followed a separate multidimensional normal distribution, the covariance matrix of which was an identity ma-

trix and the mean of which was randomly sampled from a uniform distribution $U(-20, 20)$ in each dimension. We can consider $_{10}C_2 = 45$ pairs of distributions. Next, we added some noise features sampled from $U(-20, 20)$ equal to the data range. For example, when the number of the noise features is 10, the number of all features is $2 + 10 = 12$. We calculated two lists, including a list of distances of 45 distribution pairs computed with 1-Wasserstein distance for two-dimensional data, and a list of distances of 45 distribution pairs computed with TWD for the data containing noise features. In Figure 1b, we show the Pearson correlation coefficient (PCC) between these lists when noise features were added. Figure 1b shows that PCC worsened as the number of noise features increased for both Quadtree and Clustertree, so TWD cannot approximate 1-Wasserstein distance on an original dimension well in the noisy settings due to the curse of dimensionality.

Selecting useful features is important in the high-dimensional setting with a large number of noise features as described above. We show that our proposed method can correctly select the original two-dimensional features among all features. In the following experiment, the number of noise features was 20 and the total number of all features was 22, and we let $x_0, x_1$ be the original features and $x_2, ..., x_{21}$ be the noise features. Suppose that the feature groups as prior information are given as $[(0, 1), (2, 3), ..., (20, 21)]$, where two features are given as a single group and the number of groups is 11. We learned weights using Algorithm 1 and checked the values of $\mathbf{w}^*$. As input to Algorithm 1, the data $\mathbf{X}$ were all 22-dimensional points belong-

ing to 10 distributions, the index-pair set as training data was $\Omega = \{(0,1), (0,2), ..., (8,9)\} (|\Omega| = 45)$, which was all pairs of distributions, and the hyperparameter $\eta$ was set to $0.1$ in both of Quadtree and Clustertree. We chose the small hyperparameter because we know the true feature group in the synthetic data, and it is reasonable to expect better results with the small regularization parameter, which makes the learned weights closer to a one-hot vector. As a result, for both methods, the value of the weight corresponding to the feature group $(0, 1)$ became more than $0.99$, and other weight values were less than $0.01$. These results indicate that our proposed method correctly selected the original two-dimensional features using the training data.

## 5 Real experiments

In this section, we evaluate the performance of our proposed method through word embedding-based document classification experiments on real datasets. We used five open datasets[1], including BBCSPORT, TWITTER, AMAZON, CLASSIC, and RECIPE. Each dataset has five splits of training and test data. The detail of the datasets is given in Appendix B. We used the $k$-nearest neighbor ($k$-NN) algorithm and evaluated its performance in terms of the average of the test error rate computed for each split. Moreover, we discuss the computational time of TWD-GFS and other methods with optimal transport in Appendix E.

### 5.1 Word embeddings

The performance of document classification using optimal transport depends on word embeddings, and using the appropriate word embedding is very important. The experiments in (Kusner et al., 2015) showed that the performance of WMD on each dataset differed depending on different word embedding models and training corpora. In this study, we consider using multiple word embeddings instead of single word embedding to obtain good performance across all datasets. A lot of methods to get a more useful word embedding from given word embeddings have been proposed (Yin and Schütze, 2016; Ghannay et al., 2016; Bollegala et al., 2018; Bollegala and Bao, 2018), and the concatenation of different word embeddings for each word is often used as a powerful baseline method. However, given the high dimensionality of the concatenated word vector, performance may be

degraded due to the curse of dimensionality. Hence, we can consider group feature selection with our proposed method to select important word vectors from the concatenated word vector. As a baseline method to avoid the curse of dimensionality of the concatenated word vector, we used PCA. The results of this experiment showed that our proposed method with Clustertree achieved the best performance among the methods using the concatenated word vector. The results also demonstrated that our proposed method achieved high performance across all datasets.

### 5.2 Other feature-robust methods

We additionally compare our proposed method with other feature-robust methods. We mentioned FROT (Petrovich et al., 2022) and SRW (Paty and Cuturi, 2019) as feature-robust methods. However, FROT needs about 30 times and SRW needs about 800 times more than TWD-GFS as shown in Figure 5 in Appendix E. Thus, applying FROT and SRW to document classification tasks is practically difficult. Therefore, we conducted experiments using FROT on BBCSPORT, TWITTER, and RECIPE, which are relatively small datasets. FROT is based on group feature selection, so we used concatenated word embeddings and considered five word embeddings as feature groups for FROT as well as our proposed method.

### 5.3 Experimental settings

As a model of word embeddings, we used word2vec[2] (Mikolov et al., 2013), Glove[3] (Pennington et al., 2014), and fastText[4] (Bojanowski et al., 2017; Mikolov et al., 2018). These pretrained models are publicly available, including versions trained on different corpora. In this experiment, we used five different pretrained models as shown in Table 1. All of these word vectors were 300-dimensional. Derived from the model and corpus, we denote the five word embeddings as word2vec, Glove(crawl), Glove(wiki), fastText(crawl), and fastText(wiki), respectively.

We denote the concatenated word vectors and the concatenated word vectors with PCA as Concat and Concat(PCA) respectively. Concat had 1500 dimensions and we reduced Concat(PCA) from 1500 to 300 dimensions by PCA. Before using PCA, we

---

[1] https://github.com/mkusner/wmd

[2] https://code.google.com/archive/p/word2vec/
[3] https://nlp.stanford.edu/projects/glove/
[4] https://fasttext.cc/docs/en/english-vectors.html

| Model | Corpus | Vocabulary | Vector length |
|---|---|---|---|
| word2vec | Google News | 3M | 300 |
| Glove | Common Crawl | 2.2M | 300 |
| | Wikipedia 2014 + Gigaword 5 | 400K | 300 |
| fastText | Common Crawl | 2M | 300 |
| (w/o subword) | Wikipedia 2017, UMBC webbase corpus and statmt.org news | 1M | 300 |

Table 1: Pretrained word embeddings used in the real experiment.

first standardized word vectors. We used TSW with three trees for the evaluation of baseline methods: Concat, Concat (PCA), and five word embeddings.

As the hyperparameters, we decided the number of index-pair set $\Omega$ to 10000, and the candidate values of $\eta$ to $[0.01, 0.05, 0.1, 0.5, 1.0]$. For $\Omega$, we randomly selected 10000 pairs of documents from the training dataset. Because we considered a problem setting in which we cannot use the labels of the distribution in training, we did not perform any validation for $\eta$ and instead determined the best value based on the experimental results for each $\eta$.

We implemented Quadtree, Clustertree, and TSW with reference to the public implementation[5] of Yamada et al. (2022). Also, we used the public implementation[6] of Petrovich et al. (2022) for FROT and set all hyperparameters to the default ones of this implementation. Besides, we used the public implementation[7] for $k$-NN without validation of $k$. We evaluated all methods using Intel Xeon CPU E5-2690 v4 (2.60GHz).

### 5.4 Preprocessing

In our proposed approach, the distances between points in the feature groups must be measured on the same scale. TWD-GFS learns the weights according to the magnitude of TWD calculated for each feature group and TWD is highly dependent on the scale of the distance between points, so this preprocessing step is required. In this experiment, we calculated the distances between all words and divided each word vector by the average of the distances for each word embedding. With this preprocessing, the distances between points in each word embedding have an average of 1, and their scales are aligned.

We explain other preprocessing required for using multiple word embeddings in Appendix C.

---

[5] https://github.com/oist/treeOT
[6] https://github.com/Mathux/FROT
[7] https://github.com/mkusner/wmd

### 5.5 Results

**The performance for different $\eta$** In Figure 2, we show the test error rates of TWD-GFS for different $\eta$ in Quadtree and Clustertree on the TWIT-TER, CLASSIC, and RECIPE datasets. We show the results for other datasets in Appendix D. We concluded that the best hyperparameter across all datasets was $\eta = 1.0$ for Quadtree and $\eta = 0.1$ for Clustertree, respectively. We show the mean of weights learned with the best $\eta$ on five splits for each dataset in Figure 3. Figure 2 shows that the performance with Quadtree improved as the value of $\eta$ increased. This result indicates that TWD-GFS with Quadtree performed better when each word vector was equally weighted, as shown in Figure 3a. When constructing Quadtree, the regions to be divided increase exponentially as the dimension of the original space increases. Then, each word vector is assigned to a different region, so the distance between each word on the tree is approximately equal. Therefore, the distances between distributions were approximately equal for each word embedding, and group feature selection based on distances did not work. Figure 2 also shows that the performance with Clustertree for $\eta = 0.1$ was higher than that for $\eta = 1.0$. Unlike Quadtree, this result indicates that selecting word embeddings with Clustertree, as shown in Figure 3b, performed better than treating all word embeddings equally. In addition, the test error rate for $\eta = 0.01$ was large for the four datasets except CLASSIC. The learned weights with $\eta = 0.01$ were close to the one-hot vector, so almost only one word embedding was selected, which rendered the performance unstable due to the randomness of the initialization when constructing the tree. Moreover, Figure 3b shows that Glove(crawl) and Glove(wiki) were useful word embeddings across all datasets.

**Comparison to the tree-based methods** Next, we compared the performance of TWD-GFS against the tree-based methods comprising five

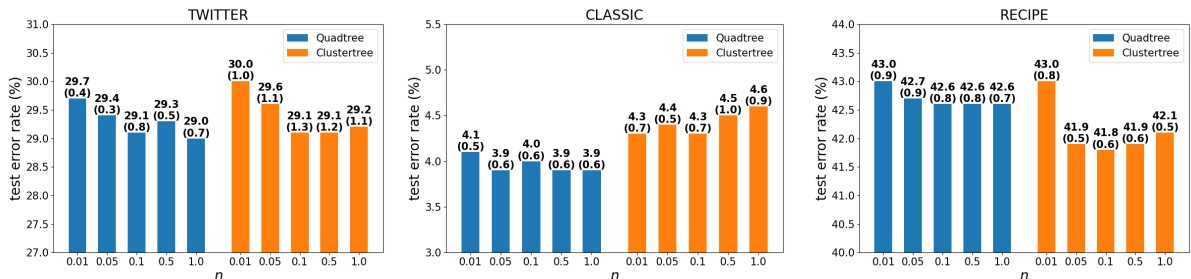

Figure 2: The test error rates in the $k$-NN of our proposed method for each $\eta$. We write the mean of five splits and the standard deviation in parentheses.

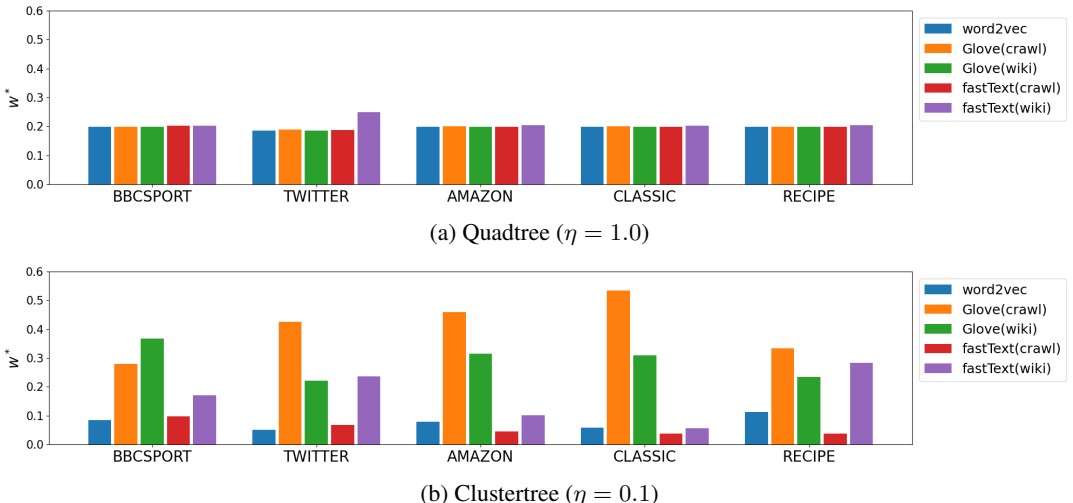

(a) Quadtree ($\eta = 1.0$)

(b) Clustertree ($\eta = 0.1$)

Figure 3: The mean of weights learned on five splits using our proposed method.

word embeddings, Concat, and Concat(PCA). The results are shown in Table 2. The performance of TWD-GFS is written only $\eta = 1.0$ for Quadtree and $\eta = 0.1$ for Clustertree. At first, we found that the best word embedding depends on the dataset and the method of tree construction, and that each word embedding performed best on at least one item. From this result, it may be expected that using these five word embeddings would perform better across the dataset. Also, we found that the performance of fastText(wiki) with Quadtree on AMAZON is considerably worse than other combinations of word embeddings of datasets. On AMAZON, the tree does not effectively approximate the original Euclidean space due to outlier words represented using fastText(wiki), making its performance worse. Using multiple word embeddings may also be expected to mitigate such challenges depending on a specific word embedding.

TWD-GFS with Quadtree did not significantly outperform Concat and Concat(PCA) because, as already mentioned, group feature selection did not work and each word embedding is equally weighted. TWD-GFS with Clustertree achieved higher performance than Concat and Concat(PCA) on all datasets. These results show that our proposed method with Clustertree can select useful features and mitigate the curse of dimensionality better than simple dimensionality reduction. Also, TWD-GFS with Clustertree achieved the best performance among all word embeddings with Clustertree on CLASSIC and RECIPE. For the other dataset, TWD-GFS with Clustertree achieved high average performance while the performance differed for each word embedding. For example with Clustertree, fastText(wiki) achieved the best performance on BBCSPORT, but performed poorly on AMAZON and CLASSIC. The performance of TWD-GFS was slightly worse than that of fastText(wiki) in BBCSPORT, but is considerably higher on AMAZON and CLASSIC.

On the other hand, the performance of TWD-GFS with Clustertree is lower than that with Quadtree on TWITTER and CLASSIC, despite Quadtree not being effective for feature selection. These results are primarily influenced by TWD's

| | Quadtree | | | | | Clustertree | | | | |
|---|---|---|---|---|---|---|---|---|---|---|
| | BBCSPORT | TWITTER | AMAZON | CLASSIC | RECIPE | BBCSPORT | TWITTER | AMAZON | CLASSIC | RECIPE |
| word2vec | **3.7 ± 0.9** | 28.6 ± 0.9 | 10.4 ± 0.4 | 4.0 ± 0.5 | 42.9 ± 1.1 | 3.3 ± 0.7 | 29.4 ± 0.9 | 11.1 ± 0.6 | 8.8 ± 1.5 | 42.5 ± 0.6 |
| Glove(crawl) | 4.2 ± 1.1 | 28.6 ± 0.5 | 10.6 ± 0.3 | **3.9 ± 0.6** | 42.7 ± 1.1 | 4.0 ± 0.8 | 29.5 ± 1.3 | **10.3 ± 0.3** | 5.1 ± 0.6 | 42.5 ± 0.6 |
| Glove(wiki) | 3.9 ± 0.7 | **28.1 ± 0.6** | **10.3 ± 0.3** | 4.0 ± 0.5 | 42.6 ± 1.0 | 3.9 ± 0.8 | **28.5 ± 1.0** | 10.4 ± 0.2 | 4.7 ± 0.5 | 42.4 ± 0.5 |
| fastText(crawl) | 3.8 ± 1.0 | **28.1 ± 0.7** | 10.4 ± 0.4 | 4.0 ± 0.6 | 42.8 ± 1.0 | 3.5 ± 0.8 | 29.5 ± 1.4 | 12.1 ± 0.8 | 6.9 ± 0.9 | 42.2 ± 0.9 |
| fastText(wiki) | 4.1 ± 0.5 | 28.7 ± 0.8 | 19.5 ± 2.4 | 4.2 ± 0.4 | 43.0 ± 0.8 | **3.2 ± 0.6** | 28.7 ± 1.1 | 12.7 ± 0.5 | 7.1 ± 0.6 | 42.4 ± 0.6 |
| Concat | 3.8 ± 0.8 | 28.6 ± 0.9 | 10.5 ± 0.4 | 4.0 ± 0.5 | **42.5 ± 1.0** | 3.8 ± 0.5 | 29.7 ± 0.9 | 10.6 ± 0.2 | 4.6 ± 0.3 | 42.2 ± 0.3 |
| Concat (PCA) | 4.2 ± 1.3 | 28.6 ± 0.8 | 10.8 ± 0.2 | 3.9 ± 0.6 | 42.6 ± 1.0 | 3.6 ± 0.8 | 29.5 ± 1.1 | 11.1 ± 0.2 | 4.8 ± 0.9 | 42.1 ± 0.3 |
| TWD-GFS($\eta = 0.1$) | - | - | - | - | - | 3.5 ± 1.2 | 29.1 ± 1.3 | 10.5 ± 0.2 | **4.3 ± 0.7** | **41.8 ± 0.6** |
| TWD-GFS($\eta = 1.0$) | 3.8 ± 1.2 | 29.0 ± 0.7 | 10.8 ± 0.7 | **3.9 ± 0.6** | 42.6 ± 0.7 | - | - | - | - | - |

Table 2: Test error rates of all tree-based methods in the $k$-NN. The lowest test error rate in each dataset is shown in bold. We write only $\eta = 1.0$ for Quadtree and $\eta = 0.1$ for Clustertree.

| | BBCSPORT | TWITTER | RECIPE |
|---|---|---|---|
| Concat | 3.8 ± 0.5 | 29.7 ± 0.9 | 42.2 ± 0.3 |
| Concat (PCA) | 3.6 ± 0.8 | 29.5 ± 1.1 | 42.1 ± 0.3 |
| TWD-GFS | 3.5 ± 1.2 | 29.1 ± 1.3 | 41.8 ± 0.6 |
| FROT | 3.6 ± 0.7 | 29.1 ± 0.5 | 41.9 ± 1.0 |

Table 3: Test error rates of FROT and three Clustertree-based methods in the $k$-NN. The error rates of Clustertree-based methods are the same as in Table 2.

inherent performance characteristics rather than the effectiveness of feature selection. Prior studies on TWD (Le et al., 2019; Takezawa et al., 2021; Yamada et al., 2022; Chen et al., 2022) have shown that the effectiveness of Quadtree and Clustertree varies depending on datasets and tasks. In our experiments on TWITTER and CLASSIC, we found that Quadtree outperforms Clustertree for other baseline methods as well as TWD-GFS. It is important to achieve higher performance than the baseline using the same tree construction method.

**Comparison to the feature-robust method** Finally, we checked the performance of FROT. We show the performance of FROT with the Clustertree-based methods in Table 3. The performance of FROT is slightly higher than Concat and Concat (PCA) with Clustertree, which are our baseline methods, and comparable to TWD-GFS with Clustertree. This result shows that the effectiveness of the feature selection of TWD-GFS with Clustertree is equivalent to FROT. We emphasize that TWD-GFS can achieve the equivalent performance of FROT with much lower computational time.

## 6 Conclusion

In this study, we proposed TWD-GFS to combine TWD and group feature selection to handle high-dimensional problems in the tree approximation of the Wasserstein distance. Unlike existing feature-robust methods of optimal transport with high computational costs, TWD-GFS can scale to large datasets owing to the rapid computation of TWD. TWD-GFS can select discriminative feature groups by solving the entropic regularized maximization problem. Through synthetic experiments, we showed that our proposed method can select discriminative features in a high-dimensional setting with a large number of noise features. Through real experiments, we considered multiple word embeddings as a feature group and applied TWD-GFS to word embedding selection. We confirmed that TWD-GFS with clustering-based trees outperformed the methods using concatenating word embeddings and achieved high performance across all datasets.

## Limitations

Our proposed approach involves two major limitations. First, TWD-GFS cannot perform group feature selection if training data is not provided. We can apply TWD-GFS to $k$-NN because training data is given, but cannot apply it to some tasks such as unsupervised semantic textual similarity (STS) tasks in which training data is not available. Second, the larger the number of feature groups, the more time is required to construct trees. In our real experiments, we used five word embedding, which did not require a large amount of time for constructing trees. If we had used one hundred word embeddings, for example, it would take 20 times longer to construct trees. When the training data is given and the number of given feature groups is not too large, TWD-GFS can be used as a fast and powerful tool for group feature selection and measuring the dissimilarity of documents.

## Acknowledgements

We thank Han Bao for his helpful comments. Sho Otao and Makoto Yamada were supported by MEXT KAKENHI Grant Number 21H04874.

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

## A  The optimal solution of (3)

In this section, we prove that the optimal solution of the constrained concave maximization problem (3) becomes (4).

*Proof.* We define $\lambda \in \mathbb{R}$ as the lagrange multiplier and define $L(\mathbf{w})$ as follows:

$$
\begin{aligned}
L(\mathbf{w}) &= \mathbf{w}^\top \boldsymbol{\phi}_\Omega + \eta H(\mathbf{w}) + \lambda(\mathbf{w}^\top \mathbf{1}_F - 1) \\
&= \mathbf{w}^\top \boldsymbol{\phi}_\Omega - \eta \sum_{f=1}^{F} w_f(\log w_f - 1) \\
&\quad + \lambda(\mathbf{w}^\top \mathbf{1}_F - 1).
\end{aligned}
$$

Taking the derivative of $L(\mathbf{w})$ with $w_f$, we get

$$
\frac{\partial L(\mathbf{w})}{\partial w_f} = (\boldsymbol{\phi}_\Omega)_f - \eta(\log w_f + 1 - 1) + \lambda.
$$

Setting $\frac{\partial L(\mathbf{w})}{\partial w_f} = 0$, we get

$$
\eta \log w_f = (\boldsymbol{\phi}_\Omega)_f + \lambda.
$$

Solving for $w_f$, we get

$$
w_f = \exp\left(\frac{(\boldsymbol{\phi}_\Omega)_f}{\eta}\right) \exp\left(\frac{\lambda}{\eta}\right).
$$

From $\mathbf{w}^\top \mathbf{1}_F = 1$, we get

$$
\exp\left(\frac{\lambda}{\eta}\right) = \frac{1}{\sum_{f=1}^{F} \exp\left(\frac{(\boldsymbol{\phi}_\Omega)_f}{\eta}\right)}.
$$

From the above results, the optimal solution $w_f^*$ becomes as follows:

$$
w_f^* = \frac{\exp\left(\frac{(\boldsymbol{\phi}_\Omega)_f}{\eta}\right)}{\sum_{k=1}^{F} \exp\left(\frac{(\boldsymbol{\phi}_\Omega)_f}{\eta}\right)}.
$$

Therefore, this is equal to (4). $\qquad\square$

## B  Dataset details

Table 4 shows the dataset details that include the number of documents, the number of unique words (bag-of-words dimension), and the average number of unique words in one document.

| name | doc num | BOW dim. | avg words |
|---|---|---|---|
| BBCSPORT | 737 | 13243 | 117 |
| TWITTER | 3108 | 6344 | 9.9 |
| AMAZON | 8000 | 42063 | 45.0 |
| CLASSIC | 7093 | 24277 | 38.6 |
| RECIPE | 4370 | 5708 | 48.5 |

Table 4: The datasets used for real experiments.

## C  Preprocessing for using multiple word embeddings

In this section, we explain two preprocessing for using multiple word embeddings in the experiments of document classification.

First, because the number of words in the vocabulary covered by each word embedding is different, we deleted words that were not covered by each word embedding and performed document classification with the deleted vocabulary. In Concat, Concat(PCA), and our proposed method, we used only the common part of the vocabulary covered by the five word embeddings and removed the other words. This preprocessing tends to disadvantage the word embeddings with a small vocabulary and our proposed method, but we confirmed that this preprocessing did not affect performance on the document classification tasks.

Second, we deleted apostrophes in a word for the word embeddings other than word2vec. The four word embeddings other than word2vec cover few words containing apostrophes. However, they cover the words with apostrophes removed in many cases. For example, "we've", a contraction of "we have", is not covered by four word embeddings other than word2vec, but "weve" without the apostrophe is covered by all word embeddings. Therefore, we deleted apostrophes in a word for the word embeddings other than word2vec to increase the available vocabulary.

## D  Additional experimental results

We show additional experimental results of document classification in Figure 4.

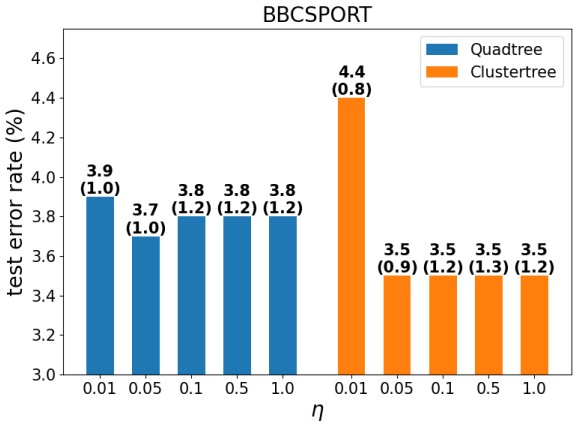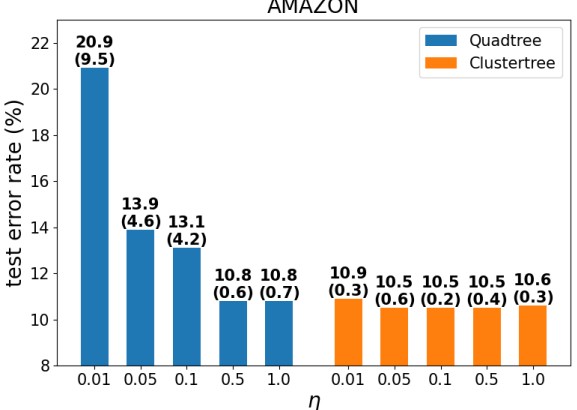

Figure 4: The test error rates in the $k$-NN of our proposed method for each $\eta$. We write the mean of five splits and the standard deviation in parentheses.

# E    Computational time

In this section, we discuss the computational time of TWD-GFS in terms of inference and training.

## E.1    Inference time

First, we discuss the computational time of measuring between distributions. We compared TWD-GFS (Quadtree and Clustertree) with Wasserstein distance solved by the Sinkhorn algorithm (Cuturi, 2013) and existing feature-robust methods that include FROT (Petrovich et al., 2022) and SRW (Paty and Cuturi, 2019). We used a matrix-form formulation proposed in (Takezawa et al., 2021) for TWD and implemented Quadtree and Clustertree with reference to the public implementation[8] of Yamada et al. (2022). We used the Python optimal transport (POT) package[9] for the Sinkhorn algorithm. We used the public implementation of Petrovich et al. (2022) for FROT[10] and Paty and Cuturi (2019) for SRW[11]. Both SRW and FROT alternately solve optimal transport problems and feature selection optimization problems, and we used the Sinkhorn algorithm as their inner solver of optimal transport problems. As a word embedding, we used 1500-dimensional vectors concatenating five word embeddings as well as used in document classification tasks. We gave five word embeddings as feature groups to TWD-GFS and FROT and set a subspace dimension of SRW to 300. We evaluated all methods on NVIDIA RTX A6000 GPU.

---

[8]https://github.com/oist/treeOT
[9]https://pythonot.github.io/index.html
[10]https://github.com/Mathux/FROT
[11]https://github.com/francoispierrepaty/SubspaceRobustWasserstein

We show the average computational time to compare 500 documents with one document for all datasets in Figure 5. We randomly sampled pairs of one document and 500 documents 50 times and measured the average computational time. We wrote a time relative to TWD-GFS with Quadtree above each bar. We show the same results separately for each method in Figure 6 to facilitate understanding of the differences for datasets in each method. Figure 5 shows that the computational time of TWD-GFS is almost the same whether Quadtree or Clustertree is used and faster than other methods on all datasets. FROT and SRW are much slower than TWD-GFS, so we confirmed that applying existing feature-robust methods with the Sinkhorn algorithm to a large dataset is difficult. In particular, SRW is very slow because it uses an eigenvalue decomposition as its inner solver for dimensionality reduction. Figure 6 shows there are slight differences in the computational time depending on datasets in each method. The computational time of TWD-GFS depends on the number of all unique words (BOW) in each dataset and that of the other methods depends on the number of unique words in a document. We confirmed the similar tendency in Figure 6. For example, because RECIPE has the smallest BOW dimension and many words in one document, TWD-GFS on RECIPE requires a low computational time and the other methods on it require a relatively high computational time.

## E.2    Training time

Second, we discuss the computational time of tree construction and training (calculating TWD) for TWD-GFS. We call Step 1 of Algorithm 1 as tree construction, and Step 2 as training. We used the

same experimental settings as in Section 5 and recorded the computational time on one of the five splits of training and test data. The results of Quadtree and Clustertree are shown in Table 5 and Table 6, respectively. The computational time for training is small because the calculation of TWD is fast. On the other hand, the computational time for tree construction seems large at first glance, but it is reasonable because we can reuse constructed trees for inference while we need to make a cost matrix for every inference in other methods.

Next, we discuss how the computational time of tree construction and training scales different datasets and feature groups. As shown in Table 5 and Table 6, the whole training time is reasonable even when dealing with large datasets, such as AMAZON. Though the cost of tree construction, which is a large part of the whole training time, depends on the BOW dimension of the dataset, we can apply our proposed approach to large datasets in practical scenarios. On the other hand, the computational time for tree construction is proportionate to the number of feature groups. Therefore, TWD-GFS may face difficulties when confronted with an extensive number of feature groups, as highlighted in the Limitations section. Exploring alternative approaches for constructing trees that are not dependent on the number of feature groups is identified as a potential avenue for future research.

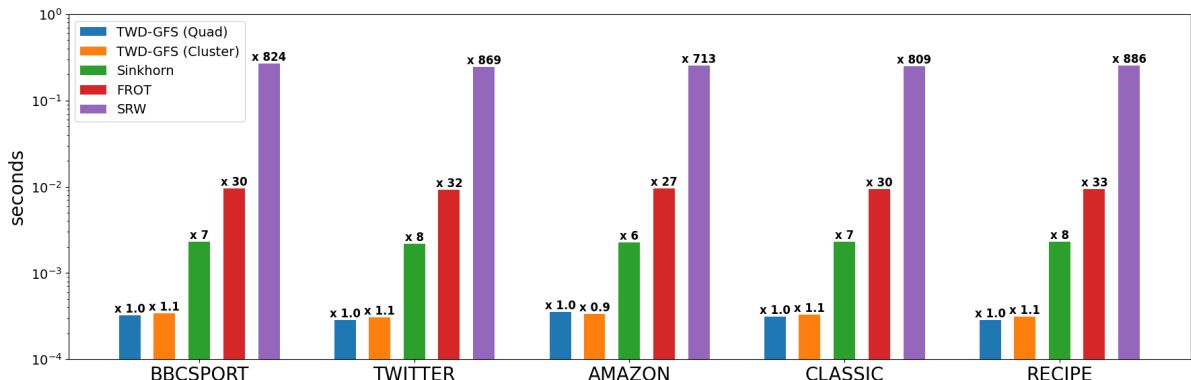

Figure 5: The average computational time to compare 500 documents with one document. The time relative to TWD-GFS with Quadtree is written above each bar.

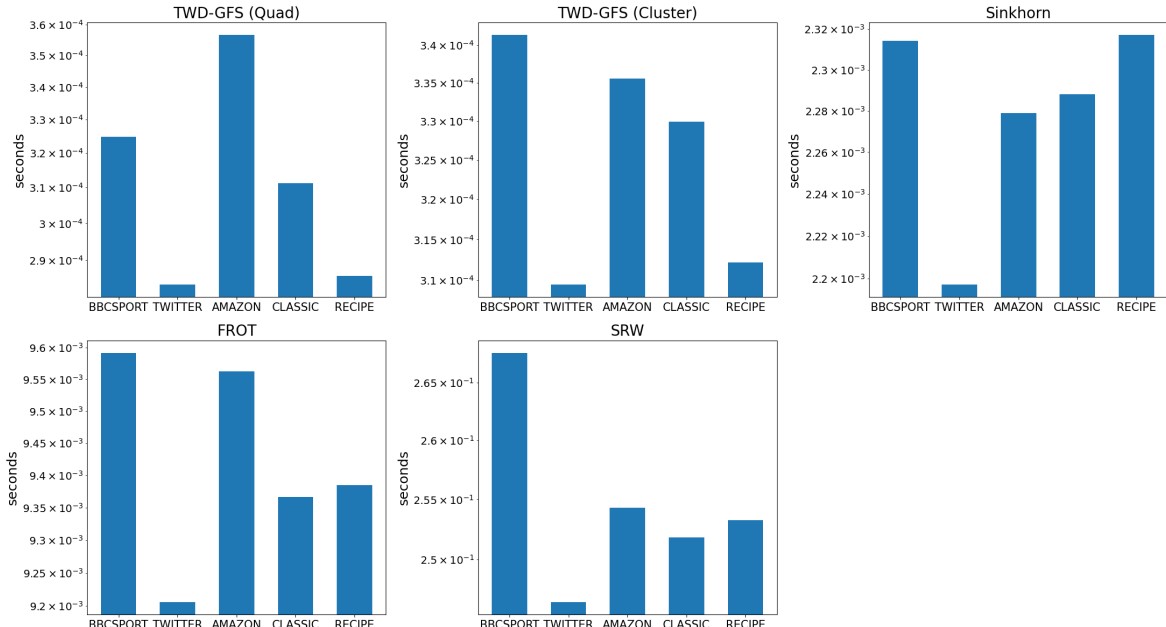

Figure 6: The average computational time to compare 500 documents with one document separately for each method. The values are the same as in Figure 5.

|                        | BBCSPORT | TWITTER | AMAZON | CLASSIC | RECIPE |
|------------------------|----------|---------|--------|---------|--------|
| Tree construction      | 82.0     | 32.7    | 293.1  | 144.0   | 36.5   |
| Training               | 4.4      | 1.8     | 3.1    | 2.6     | 3.0    |
| Tree const. + Training | 86.4     | 34.5    | 296.2  | 146.6   | 39.5   |

Table 5: The computational time [seconds] for tree construction and training for Quadtree.

|                        | BBCSPORT | TWITTER | AMAZON | CLASSIC | RECIPE |
|------------------------|----------|---------|--------|---------|--------|
| Tree construction      | 48.4     | 23.4    | 139.3  | 84.2    | 25.1   |
| Training               | 4.6      | 1.8     | 3.1    | 2.6     | 2.7    |
| Tree const. + Training | 53.0     | 25.2    | 142.4  | 86.8    | 27.8   |

Table 6: The computational time [seconds] for tree construction and training for Clustertree.