# OpenReview forum: "A linear time approximation of Wasserstein distance with word embedding selection"
_EMNLP/2023/Conference — EMNLP 2023 Main_

### Official Review · Reviewer_2jWw · 2023-08-04

**Soundness:** 3

**Excitement:**

4: Strong: This paper deepens the understanding of some phenomenon or lowers the barriers to an existing research direction.

**Paper Topic And Main Contributions:**

This paper presents a TWD-based method which automatically selects useful word embeddins during a tree approximation of Wasserstein distance.

**Questions For The Authors:**

Question A: Why different baselines are used in the effectiveness and efficiency testing? It is better to list the performance of FROT and SRW in Table 2.

Question B: Explain why TWD-GFS with Clustertree performs worse than TWD-GFS with Quadtree?

**Reasons To Accept:**

The TWD-based method solves the discriminative feature group selection problem during a tree approximation of Wasserstein distance, and the proposed method has been proved be effective and efficient.

**Reasons To Reject:**

The experimental result is not very convincing.

1)	Different baselines are used in the effectiveness and efficiency testing. For the effectiveness testing, Concat and Concat (PCA) are used. For the efficiency testing, FROT, SRW, and Sinkhorn are chosen.

2)	As shown in Table 2, although TWD-GFS with Clustertree significantly outperform Concat and Concat(PCA), the overall performance of TWD-GFS with Clustertree is worse than TWD-GFS with Quadtree. This indicates TWD-GFS is applied effectively only to some tree approximation methods.

**Reproducibility:**

3: Could reproduce the results with some difficulty. The settings of parameters are underspecified or subjectively determined; the training/evaluation data are not widely available.

**Reviewer Confidence:**

3: Pretty sure, but there's a chance I missed something. Although I have a good feel for this area in general, I did not carefully check the paper's details, e.g., the math, experimental design, or novelty.

---

> ### Author Rebuttal · Authors · 2023-08-28
>
> Thank you for your valuable questions and comments.
>
> > Question A: Why different baselines are used in the effectiveness and efficiency testing? It is better to list the performance of FROT and SRW in Table 2.
>
> FROT needs about $30$ times and SRW needs about $800$ times more than TWD-GFS as shown in Figure 6, so it is practically difficult to apply them to document classification tasks with large datasets. Thus, we could not include those results in the submission.
>
> As a reference, we conducted new document classification experiments using FROT for BBCSPORT, TWITTER, RECIPE and show their results in the following table. We used concatenated word embeddings and considered five word embeddings as feature groups as well as TWD-GFS. The performance of FROT is slightly higher than our baseline, Concat and Concat with PCA, and comparable to TWD-GFS with Clustertree, which is our most important results. We emphasize that TWD-GFS can achieve the equivalent performance with FROT with much lower computation time than FROT. We will include these results in the final version.
>
> |   the error rate [%]              | BBCSPORT      | TWITTER        | RECIPE        |
> |-----------------------------------|---------------|----------------|----------------|
> | Concat (Clustertree)              | $3.8 \pm 0.5$ | $29.7 \pm 0.9$ | $42.2 \pm 0.3$ |
> | Concat with PCA (Clustertree)     | $3.6 \pm 0.8$ | $29.5 \pm 1.1$ | $42.1 \pm 0.3$ |
> | TWD-GFS (Clustertree, $\eta=0.1$) | $3.5 \pm 1.2$ | $29.1 \pm 1.3$ | $41.8 \pm 0.6$ |
> | FROT                              | $3.6 \pm 0.7$ | $29.1 \pm 0.5$ | $41.9 \pm 1.0$ |
>
> > Question B: Explain why TWD-GFS with Clustertree performs worse than TWD-GFS with Quadtree?
>
> These results are influenced by TWD's inherent performance characteristics, rather than our method itself. Prior studies on TWD [1, 2, 3, 4] illustrate that Quadtree or Clustertree's effectiveness varies based on dataset and task dynamics. This TWD trait significantly shapes our experiments. For instance, Quadtree is more effective than Clustertree in TWITTER and CLASSIC datasets with TWD-GFS as well as other baselines. Consequently, TWD-GFS's performance with Clustertree lags behind that with Quadtree, despite optimal feature group selection. It is important to achieve higher performance than the baseline using the same tree-construction method.
>
> These answers can also address the concern you mentioned in the "Reasons To Reject" section.
>
> ## References
>
> [1] Tam Le, Makoto Yamada, Kenji Fukumizu, and Marco Cuturi. 2019. Tree-sliced variants of wasserstein distances. In NeurIPS.
>
> [2] Yuki Takezawa, Ryoma Sato, and Makoto Yamada. 2021. Supervised tree-wasserstein distance. In ICML.
>
> [3] Makoto Yamada, Yuki Takezawa, Ryoma Sato, Han Bao, Zornitsa Kozareva, and Sujith Ravi. 2022. Approximating 1-wasserstein distance with trees. Transactions on Machine Learning Research.
>
> [4] Samantha Chen, Puoya Tabaghi, and Yusu Wang. 2022. Learning ultrametric trees for optimal transport regression. arXiv preprint arXiv:2210.12288.

---

### Official Review · Reviewer_yRiE · 2023-08-04

**Soundness:** 3

**Excitement:**

3: Ambivalent: It has merits (e.g., it reports state-of-the-art results, the idea is nice), but there are key weaknesses (e.g., it describes incremental work), and it can significantly benefit from another round of revision. However, I won't object to accepting it if my co-reviewers champion it.

**Paper Topic And Main Contributions:**

This paper proposes to apply dimensional reduction techniques on the tree-Wasserstein metric in the application of document distance. The key technique is to softly select the best-performing feature group based on the observed training data.

**Questions For The Authors:**

1. In the synthetic experiments, how do we set up feature groups?

**Reasons To Accept:**

1. The goal of this paper is necessary when the TWD losses its approximation property gradually in high dimensional space. An entropy algorithm is proposed

**Reasons To Reject:**

1. The technical contribution is limited. The key idea of this feature selection is to identify a soft maximum of tree Wasserstein distances conditional on groups. It relies on a pre-defined group. So this restricts the application methods in such a specific case where multiple feature groups are available.
2. There are still gaps between the performance of the targeting task and the feature selection problem. If this method is proposed for document distance style application, it is not clear why a better feature selection yields better performance of document distance.
3. The baseline is lacking, especially for the dimensional reduction methods in optimal transport estimation. For example, SRW and FROT, as mentioned in this paper, should be included as baselines.

**Reproducibility:**

4: Could mostly reproduce the results, but there may be some variation because of sample variance or minor variations in their interpretation of the protocol or method.

**Reviewer Confidence:**

4: Quite sure. I tried to check the important points carefully. It's unlikely, though conceivable, that I missed something that should affect my ratings.

---

> ### Author Rebuttal · Authors · 2023-08-28
>
> Thank you for your valuable questions and comments.
>
> > 1. The technical contribution is limited. The key idea of this feature selection is to identify a soft maximum of tree Wasserstein distances conditional on groups. It relies on a pre-defined group. So this restricts the application methods in such a specific case where multiple feature groups are available.
>
> We respectfully disagree with this comment. In feature selection literatures, having a pre-defined group is a common setting and useful in practice. For example, the group Lasso [1], which is a widely used statistics method, assumes to have a pre-defined feature group. Moreover, we can use the proposed method for a standard feature selection setup if we do not have prior information; each group consists of a feature.
>
> Tree-Wasserstein distance is renowned for its speed and performance in document distance style application, relying heavily on word embeddings. Furthermore, leveraging multiple embeddings enhances performance across diverse NLP tasks [2, 3, 4, 5]. However, integrating multiple word embeddings can introduce the curse of dimensionality due to their high dimensionality. Our proposed approach effectively addresses this challenge.
>
>
> > 2. There are still gaps between the performance of the targeting task and the feature selection problem. If this method is proposed for document distance style application, it is not clear why a better feature selection yields better performance of document distance.
>
> In general, since the tree-Wasserstein distance computation heavily depends on word embeddings as compared in Table 2, the document classification performance can be improved by selecting a good word embedding for the task. Thus, if we can automatically select useful word embeddings, we can improve the classification accuracy of document classification tasks.
>
> We assume that the reviewer wonder why we can have performance improvement by an unsupervised feature selection approach without document labels over the document classification tasks. In our proposed method, we assume that the tree-Wasserstein distances that has larger value are more informative. FROT and SRW also assume this assumption. Note that, as an extreme case, if the tree-Wasserstein distance has exact same value for all documents, the classification accuracy is zero. Thus, selecting a feature maximizing the tree-Wasserstein distance is a reasonable approach for unsupervised tasks, and this assumption is well supported from the experiments.
>
>
> > 3. The baseline is lacking, especially for the dimensional reduction methods in optimal transport estimation. For example, SRW and FROT, as mentioned in this paper, should be included as baselines.
>
> FROT needs about $30$ times and SRW needs about $800$ times more than TWD-GFS as shown in Figure 6, so it is practically difficult to apply them to document classification tasks with large datasets. Thus, we could not include those results in the submission.
>
> As a reference, we conducted new document classification experiments using FROT for BBCSPORT, TWITTER, RECIPE and show their results in the following table. We used concatenated word embeddings and considered five word embeddings as feature groups as well as TWD-GFS. The performance of FROT is slightly higher than our baseline, Concat and Concat with PCA, and comparable to TWD-GFS with Clustertree, which is our most important results. We emphasize that TWD-GFS can achieve the equivalent performance with FROT with much lower computation time than FROT. We will include these results in the final version.
>
> |   the error rate [%]              | BBCSPORT      | TWITTER        | RECIPE        |
> |-----------------------------------|---------------|----------------|----------------|
> | Concat (Clustertree)              | $3.8 \pm 0.5$ | $29.7 \pm 0.9$ | $42.2 \pm 0.3$ |
> | Concat with PCA (Clustertree)     | $3.6 \pm 0.8$ | $29.5 \pm 1.1$ | $42.1 \pm 0.3$ |
> | TWD-GFS (Clustertree, $\eta=0.1$) | $3.5 \pm 1.2$ | $29.1 \pm 1.3$ | $41.8 \pm 0.6$ |
> | FROT                              | $3.6 \pm 0.7$ | $29.1 \pm 0.5$ | $41.9 \pm 1.0$ |
>
> > 1. In the synthetic experiments, how do we set up feature groups?
>
> In our synthetic experiments, we crafted 22-dimensional synthetic data, denoted as $\mathbf{x}=(x_0, x_1, ..., x_{21})$. Within this setup, $x_0$ and $x_1$ are the correct features, while $x_2, ..., x_{21}$ are the noise features.
>
> To evaluate the efficacy of TWD-GFS in selecting the appropriate feature groups, we segmented the features into "correct" and "noise" groups. This enabled us to assess TWD-GFS's capacity to identify the correct feature groups effectively. The simplest case involved two groups, namely $(0, 1)$ for correct features and $(2, 3, ..., 21)$ for noise features. In our synthetic experiments, we introduced a slightly more complex scenario, encompassing multiple noise feature groups, defined as $(0, 1), (2, 3), ..., (20, 21)$.
>
> ## References
>
> [1] Ming Yuan and Yi Lin. Model selection and estimation in regression with grouped variables. 2006. Journal of the Royal Statistical Society: Series B (Statistical Methodology), 68(1):49–67
>
> [2] Wenpeng Yin and Hinrich Schütze. 2016. Learning word meta-embeddings. In Proceedings of the 54th Annual Meeting of the Association for Computational Linguistics.
>
> [3] Sahar Ghannay, Benoit Favre, Yannick Estève, and Nathalie Camelin. 2016. Word embedding evaluation and combination. In Proceedings of the Tenth International Conference on Language Resources and Evaluation (LREC’16)
>
> [4] Danushka Bollegala, Kohei Hayashi, and Ken-ichi Kawarabayashi. 2018. Think globally, embed locally—locally linear meta-embedding of words. In IJCAI.
>
> [5] Danushka Bollegala and Cong Bao. 2018. Learning word meta-embeddings by autoencoding. In Proceedings of the 27th International Conference on Computational Linguistics.

---

### Official Review · Reviewer_ybsA · 2023-08-12

**Soundness:** 3

**Excitement:**

4: Strong: This paper deepens the understanding of some phenomenon or lowers the barriers to an existing research direction.

**Paper Topic And Main Contributions:**

This paper studies the calculation of the word-mover's distance with word embeddings and proposes an approximation-based approach to improve performance. It presents a tree-based method to measure inter-distribution distances which can select discriminative features in a high-dimensional and noisy setting and manage to achieve good performance.

**Questions For The Authors:**

A. Line 407: Why were these values of eta chosen for synthetic data? The values appear to be opposite to the optimal ones for real data.
B. Figure 3: Why is the performance of fastText(wiki) on Twitter considerably worse than all other embedding/dataset combinations?
C. Lines 569-480: Do you have any intuition for the reasons behind these results?
D. Line 791: Why not compare tree-construction and training time for TWD-GFS with cost matrix calculation time for other methods? How did that scale for different datasets and feature groups?

**Reasons To Accept:**

Calculating the word-mover's distance is a useful metric for comparing the similarity of text documents, and this paper presents a way to speed up these comparisons when working with large sets of documents. The performance gain over prior approaches is substantial.

**Reasons To Reject:**

No strong reasons to reject. The approach appears to work well primarily for a select range of training data size and feature group numbers, as is addressed in the limitations section.

**Reproducibility:**

4: Could mostly reproduce the results, but there may be some variation because of sample variance or minor variations in their interpretation of the protocol or method.

**Reviewer Confidence:**

2: Willing to defend my evaluation, but it is fairly likely that I missed some details, didn't understand some central points, or can't be sure about the novelty of the work.

**Typos Grammar Style And Presentation Improvements:**

Line 406: Remove extra space before comma.
Line 418: Remove extra space before footnote.
Line 807: Missing word? “As a word embedding, we used 1500-dimensionnal vectors concatenating five word embeddings as well as used in document classification tasks.”
Line 465: Should be “... on different corpora”.
Figure 7: Y-axes should have the same scale for direct comparison.

---

> ### Author Rebuttal · Authors · 2023-08-28
>
> Thank you for your valuable questions and comments.
>
> > A. Line 407: Why were these values of eta chosen for synthetic data? The values appear to be opposite to the optimal ones for real data.
>
> $\eta$ can control how soft the feature weights are. If we set the regularization smaller, a few features are used for Wasserstein computation, while the large number of features can be used if we set the large regularization parameter.
>
> In the synthetic data, we know the true feature set, while we do not know the true feature set for real-world data. Thus, it is natural to get better results with the small regularization parameter for synthetic data, while we may get good performance with the large regularization parameter for real-world data. In our experiment, we set $\eta$ to $1.0$ in Clustertree, because we observed that $\eta=1.0$ makes the feature weight vector to a one-hot vector. It does not change the result even if we use $\eta = 0.1$. In the final version, to avoid confusion, we will change the result with $\eta=0.1$.
>
>
> > B. Figure 3: Why is the performance of fastText(wiki) on Twitter considerably worse than all other embedding/dataset combinations?
>
> It seems that you might be referring to Table 2 instead of Figure 3, and AMAZON instead of TWITTER.
> In AMAZON, the tree does not effectively approximate the original Euclidean space due to outlier words represented using fastText(wiki), making its performance worse. Using multiple word embeddings can help mitigate such challenges depending on a specific word embedding.
>
> > C. Lines 569-480: Do you have any intuition for the reasons behind these results?
>
> It has been reported that using multiple word embeddings improves the average performance for many datasets in many NLP tasks [1, 2, 3, 4]. So, it is natural that TWD with multiple word embeddings can get high performance. Moreover, TWD-GFS can achieve higher accuracy because it can avoid the curse of dimensionality thanks to the feature selection.
>
> > D. Line 791: Why not compare tree-construction and training time for TWD-GFS with cost matrix calculation time for other methods?
>
> Thank you for pointing this out. In this paper, we focus more on the computation in inference time. Overall, the computation of the training time for Quadtree and Clustertree are reasonable as summarized in the following tables. Note that we need to construct trees only once and can reuse for inference time while we need to make a cost matrix for every inference in other methods. Moreover, the inference time for the proposed method is much smaller than that of Sinkhorn, FROT and SRW as shown in Figure 6 of the appendix.
>
> |   Quadtree  [seconds]       | BBCSPORT | TWITTER | AMAZON | CLASSIC | RECIPE |
> |-----------------------------|----------|---------|--------|---------|---------|
> | Tree construction ($T=5$)   | 82.0     | 32.7    | 293.1  | 144.0   | 36.5    |
> | Training (\|$\Omega$\|=10000) | 4.4      | 1.8     | 3.1    | 2.6     | 3.0     |
> | Tree const. + Training      | 86.4     | 34.5    | 296.2  | 146.6   | 39.5    |
>
> |   Clustertree  [seconds]    | BBCSPORT | TWITTER | AMAZON | CLASSIC | RECIPE |
> |-----------------------------|----------|---------|--------|---------|---------|
> | Tree construction ($T=5$)   | 48.4     | 23.4    | 139.3  | 84.2    | 25.1    |
> | Training (\|$\Omega$\|=10000) | 4.6      | 1.8     | 3.1    | 2.6     | 2.7     |
> | Tree const. + Training      | 53.0     | 25.2    | 142.4  | 86.8    | 27.8    |
>
> >  How did that scale for different dataset and feature groups?
>
> As shown in the above table, the training time of TWD-GFS is reasonable even when dealing with large datasets, such as AMAZON.
> Though the cost of tree-construction, which is a large part of training time, depends on the BOW dimension of dataset, we can apply our proposed approach to large datasets in practical scenarios.
>
> On the other hand, the computational time for tree construction is proportionate to the number of feature groups. So, TWD-GFS may face difficulties when confronted with an extensive number of feature groups, as highlighted in the Limitations section. Exploring alternative approaches for constructing trees that are not solely dependent on the number of feature groups is identified as a potential avenue for future research.
>
> We will include the discussion in the final version.
>
>
>
> > Figure 7: Y-axes should have the same scale for direct comparison.
>
> We have already reported the results with the same time scale in Figure 6.
>
> ## References
>
> [1] Wenpeng Yin and Hinrich Schütze. 2016. Learning word meta-embeddings. In Proceedings of the 54th Annual Meeting of the Association for Computational Linguistics.
>
> [2] Sahar Ghannay, Benoit Favre, Yannick Estève, and Nathalie Camelin. 2016. Word embedding evaluation and combination. In Proceedings of the Tenth International Conference on Language Resources and Evaluation (LREC’16)
>
> [3] Danushka Bollegala, Kohei Hayashi, and Ken-ichi Kawarabayashi. 2018. Think globally, embed locally—locally linear meta-embedding of words. In IJCAI.
>
> [4] Danushka Bollegala and Cong Bao. 2018. Learning word meta-embeddings by autoencoding. In Proceedings of the 27th International Conference on Computational Linguistics.

---

### Meta-Review · Area_Chair_JLzf · 2023-09-19

**Recommendation:** 4

**Metareview:**

Wasserstein distance is an attractive method for comparing probability distributions because it can easily be applied in NLP for tasks involving comparing documents (eg, classification). Its computational cost remains a bottleneck and various methods have been proposed to accelerate it. This paper proposes an improvement over one of those methods (tree Wasserstein distance), which is beneficial when applied to high-dimensional problems (as is normally the case in NLP).

Reviewers pointed out the originality of the approach as well as the empirical results (gains are substantial) as strengths. There were lingering questions around some experimental decisions, in particular regarding the use of different baselines for different datasets and the relative performance of some proposed combinations. The authors addressed those questions - their presence highlighted by various reviewers seem to indicate that those are point that should be clarified in the manuscript.

---

### Decision · Program_Chairs · 2023-10-07

**Decision:**

Accept-Main

**Comment:**

Wasserstein distance is an attractive method for comparing probability distributions because it can easily be applied in NLP for tasks involving comparing documents (eg, classification). Its computational cost remains a bottleneck and various methods have been proposed to accelerate it. This paper proposes an improvement over one of those methods (tree Wasserstein distance), which is beneficial when applied to high-dimensional problems (as is normally the case in NLP).

Reviewers pointed out the originality of the approach as well as the empirical results (gains are substantial) as strengths. There were lingering questions around some experimental decisions, in particular regarding the use of different baselines for different datasets and the relative performance of some proposed combinations. The authors addressed those questions - their presence highlighted by various reviewers seem to indicate that those are point that should be clarified in the manuscript.